# Practical diagnosis of cirrhosis in non-alcoholic fatty liver disease using currently available non-invasive fibrosis tests

Jérôme Boursier [1,2] ✉, Marine Roux[2], Charlotte Costentin[3,4], Julien Chaigneau[2], Céline Fournier-Poizat[5], Aldo Trylesinski[6], Clémence M. Canivet[1,2], Sophie Michalak[2,7], Brigitte Le Bail[8,9], Valérie Paradis[10], Pierre Bedossa[10,11], Nathalie Sturm[12], Victor de Ledinghen[9,13], AFEF group for the study of liver fibrosis*, M118 study group* & Philip N. Newsome [14,15,16]

Unlike for advanced liver fibrosis, the practical rules for the early non-invasive diagnosis of cirrhosis in NAFLD remain not well defined. Here, we report the derivation and validation of a stepwise diagnostic algorithm in 1568 patients with NAFLD and liver biopsy coming from four independent cohorts. The study algorithm, using first the elastography-based tests Agile3+ and Agile4 and then the specialized blood tests FibroMeter[V3G] and CirrhoMeter[V3G], provides stratification in four groups, the last of which is enriched in cirrhosis (71% prevalence in the validation set). A risk prediction chart is also derived to allow estimation of the individual probability of cirrhosis. The predicted risk shows excellent calibration in the validation set, and mean difference with perfect prediction is only −2.9%. These tools improve the personalized non-invasive diagnosis of cirrhosis in NAFLD.

The prognosis for patients with non-alcoholic fatty liver disease (NAFLD), the most prevalent cause of chronic liver disease worldwide, is closely linked to their stage of liver fibrosis[1,2]. Several non-invasive strategies, mainly utilising blood tests and elastography devices, are now available to identify the subset of at-risk patients in clinical practice[3]. Algorithms combining non-invasive tests have been proposed, not only to improve the diagnostic accuracy but also to define the optimal sequence of usage to identify patients who need referral to the liver specialist[4,5]. In its latest 2021 guidelines, the European Association for the Study of the Liver (EASL) has proposed a three-tiered

algorithm for the diagnosis of advanced liver fibrosis in NAFLD using first the simple blood test FIB4, then liver stiffness measurement with vibration-controlled transient elastography (VCTE), and finally patented blood tests[6]. As it enables a diagnosis of advanced liver fibrosis without liver biopsy when all the three diagnostic lines agree, this algorithm is innovative and represents a significant advancement in the management of patients with NAFLD.

Identifying patients with cirrhosis non-invasively, as opposed to advanced fibrosis, is a major priority in the field of NAFLD, as such patients require specific clinical management including screening for

[1]Hepato-Gastroenterology and Digestive Oncology Department, Angers University Hospital, Angers, France. [2]HIFIH Laboratory, SFR ICAT 4208, Angers University, Angers, France. [3]Univ. Grenoble Alpes, Clinique Universitaire d'Hépato-gastroentérologie, CHU Grenoble Alpes, Grenoble, France. [4]Univ. Grenoble Alpes; Institute for Advanced Biosciences, CNRS UMR 5309-INSERM U1209, Grenoble, France. [5]Echosens, Paris, France. [6]AdvanzPharma, London, UK. [7]Pathology Department, Angers University Hospital, Angers, France. [8]Pathology Department, Bordeaux University Hospital, Bordeaux, France. [9]Bordeaux Institute of Oncology, BRIC UMR U1312, INSERM, Université de Bordeaux, Bordeaux, France. [10]Department of Pathology, Physiology and Imaging, Beaujon Hospital Paris Diderot University, Paris, France. [11]Liverpat, Paris, France. [12]Pathology Department, CHU Grenoble Alpes, Grenoble, France. [13]Hepatology Unit, Haut Leveque hospital, Bordeaux University Hospital, Bordeaux, France. [14]National Institute for Health Research, Birmingham Biomedical Research Centre at University Hospitals Birmingham NHS Foundation Trust, Birmingham, UK. [15]Centre for Liver & Gastrointestinal Research, Institute of Immunology and Immunotherapy, University of Birmingham, Birmingham, UK. [16]Liver Unit, University Hospitals Birmingham NHS Foundation Trust, Birmingham, UK. *Lists of authors and their affiliations appears at the end of the paper. ✉e-mail: jerome.boursier@univ-angers.fr

hepatocellular carcinoma, oesophageal varices and sarcopaenia as recommended by most international guidelines[7–9]. Moreover, specific therapeutic trials are being conducted for patients with cirrhotic NASH[10], making it necessary to accurately select candidates for these studies. New non-invasive tests specifically developed for the diagnosis of cirrhosis are now available. Agile4 combines VCTE results with routine parameters from liver evaluation (serum transaminases, platelets, with sex and diabetes) in a formula dedicated to cirrhosis[11]. CirrhoMeter[V3G] (CMV3G) has been developed to target cirrhosis by using the same biomarkers than those included in the specialized blood test FibroMeter[V3G] (FMV3G)[12]. These tests dedicated to the diagnosis of cirrhosis offer the opportunity to improve the currently recommended algorithms for the non-invasive diagnosis of advanced liver fibrosis. Indeed, calculated at the same time as the tests used in these algorithms without the need for additional biomarkers, they may contribute to discriminate a new group including cirrhotic patients.

In this work, we used the best performing non-invasive tests currently available to liver specialists to develop and validate an accurate diagnosis of cirrhosis in patients with NAFLD, which translated in a "cirrhosis" category in the diagnostic algorithms currently recommended.

## Results

### Patients

From the 1757 patients initially available, 1568 were included in the study (see the flow-chart of the study in Supplementary Fig. s1). Their characteristics are summarized in Table 1. Median age was 57.6 years, 59.1% of the patients were male, median BMI was 31.6 kg/m², and 51.3% had type 2 diabetes mellitus. The median biopsy length was 25 mm, and 91.6% of the biopsies were at least 15 mm in length. The prevalence of advanced liver fibrosis F34 was 38.3%, and cirrhosis 12.1%.

### Accuracy of non-invasive tests

Our results confirmed the very good to excellent areas under the receiver operating characteristic (AUROC) of non-invasive tests for the diagnosis of cirrhosis with results reaching 0.90 for the best tests (Table 2). For both advanced fibrosis and cirrhosis, the NAFLFD fibrosis score was the least accurate fibrosis test and elastography-based tests performed better than blood fibrosis tests (Table 2).

Beyond AUROC analysis, Supplementary Fig. s2 shows that the presence of cirrhosis progressively increased with elevations of fibrosis tests results. We therefore evaluated the accuracy of rule-out/rule-in thresholds for tests dedicated to the diagnosis of cirrhosis, alongside those tests focussed on the diagnosis of advanced fibrosis (see respectively F4 and F34 thresholds in Supplementary Table s1). Diagnostic accuracy of these thresholds was consistent between the derivation and the validation sets (Supplementary Figs. s3, 4 and Supplementary Tables s2, 3). To rule out cirrhosis, the low F34 threshold performed better than the low F4 threshold with a two to threefold decrease in the number of false negatives (Supplementary Figs. s3, 4), resulting in an increase in sensitivity to > 95% (Supplementary Tables s2, 3). On the opposite, the high F4 threshold performed better than the high F34 threshold for ruling in cirrhosis, with a 2 to 6-fold decrease in the number of false positives, and higher specificity that reached > 92%. Taken together, these results suggested that combining the low F34 threshold with the high F4 threshold was the best approach to respectively rule out and rule in cirrhosis.

### Fibrosis test complementarity

Fibrosis tests from each respective couple, FMV3G/CMV3G and Agile3 + /Agile4, include the same variables (Supplementary Table s1), albeit with one test dedicated to the diagnosis of advanced liver fibrosis (FMV3G, Agile3 + ) and the other on the diagnosis of cirrhosis (CMV3G, Agile4). We therefore evaluated the complementarity of the tests from each of these two couples. Agile3+ and Agile4 were very well correlated, and the relation between these two tests was exponentially shaped (Supplementary Fig. s5). Therefore, by cross linking the three intervals defined by the F34 thresholds of Agile3+ with the three intervals defined by the F4 thresholds of Agile4, we observed that Agile4 stratified the risk of cirrhosis within the higher interval of Agile3+ (Fig. 1a). Indeed, in patients with Agile3 + ≥0.679, the prevalence of cirrhosis was even higher when Agile4 was > 0.474. This led us to develop the Agile3 + /4 classification including four groups with increasing prevalence of advanced liver fibrosis and cirrhosis (Fig. 1b): (i) Agile3 + <0.451; (ii) Agile3+ between 0.451 and 0.678; (iii) Agile

## Table 1 | Patient characteristics

| | All (n = 1568) | Derivation set (n = 872) | Validation set (n = 696) | p |
|---|---|---|---|---|
| Age (years) | 57.6 [48.6;64.9] | 57.0 [48.0;64.0] | 58.4 [49.4;65.7] | 0.020 |
| Male sex (%) | 59.1 | 56.4 | 62.5 | 0.015 |
| BMI (kg/m²) | 31.6 [28.3;36.3] | 31.9 [28.1;36.8] | 31.6 [28.5;36.0] | 0.776 |
| T2DM (%) | 51.3 | 52.5 | 49.9 | 0.309 |
| Biopsy length (mm) | 25 [20;33] | 24 [18;30] | 30 [22;35] | 1.1e⁻¹⁹ |
| NAFLD Activity Score | 4 [3;5] | 5[3;6] | 4 [3;5] | 1.1e⁻⁹ |
| Fibrosis stage (%): | | | | 0.015 |
| - F0 | 12.1 | 12.4 | 11.6 | |
| - F1 | 22.6 | 23.4 | 21.6 | |
| - F2 | 27.1 | 24.2 | 30.7 | |
| - F3 | 26.2 | 26.1 | 26.3 | |
| - F4 | 12.1 | 13.9 | 9.8 | |
| Fibrosis F34 (%) | 38.3 | 40.0 | 36.1 | 0.117 |
| Cirrhosis F4 (%) | 12.1 | 13.9 | 9.8 | 0.015 |
| AST (IU/l) | 39 [29;55] | 39 [30;56] | 38 [28;54] | 0.137 |
| ALT (IU/l) | 55 [36;79] | 54 [35;79] | 56 [37;82] | 0.244 |
| Gamma-GT (IU/l) | 72 [40;139] | 71 [41;141] | 73 [40;133] | 0.759 |
| Bilirubin (µmol/l) | 10 [7;13] | 10 [7;13] | 10 [7;14] | 0.715 |
| Albumin (g/l) | 43.4 [41.0;46.0] | 43.4 [41.1;46.0] | 43.9 [41.0;46.0] | 0.569 |
| Platelets (G/l) | 222 [181;265] | 224 [183;266] | 218 [179;261] | 0.163 |
| Prothrombin time (%) | 95 [88;101] | 93 [86;100] | 97 [90;103] | 1.9e⁻¹¹ |
| NAFLD fibrosis score | −0.801 [−1.936;0.279] | −0.800 [−1.974;0.300] | −0.801 [−1.885;0.218] | 0.704 |
| FIB4 | 1.36 [0.92;2.04] | 1.34 [0.91;2.07] | 1.37 [0.93;2.00] | 0.915 |
| FibroMeter[V3G] | 0.48 [0.28;0.73] | 0.47 [0.28;0.74] | 0.48 [0.28;0.70] | 0.219 |
| CirrhoMeter[V3G] | 0.02 [0.01;0.10] | 0.02 [0.01;0.10] | 0.02 [0.01;0.09] | 0.105 |
| VCTE (kPa) | 8.7 [6.1;13.2] | 8.5 [6.1;12.5] | 8.8 [6.1;13.9] | 0.213 |
| Agile3+ | 0.43 [0.14;0.77] | 0.43 [0.13;0.77] | 0.43 [0.14;0.76] | 0.826 |
| Agile4 | 0.04 [0.01;0.17] | 0.05 [0.01;0.19] | 0.04 [0.01;0.16] | 0.869 |

*ALT* Alanine aminotransferase, *AST* Aspartate aminotransferase, *BMI* Body mass index, *NAFLD* Non alcoholic fatty liver disease, *T2DM* Type 2 diabetes mellitus, *VCTE* Vibration-controlled transient elastography.
Continuous variables were expressed as median with first and third quartiles and compared using the Mann-Whitney test or the Kruskal-Wallis test when appropriate. Categorical variables were expressed as percentages and compared using the Chi-squared test or the Fisher test when appropriate. All statistical tests were two-sided.

**Table 2 | AUROC of non-invasive tests for advanced liver fibrosis and cirrhosis**

| Diagnostic target | Fibrosis test | All (n = 1568) | Derivation set (n = 872) | Validation set (n = 696) | p |
|---|---|---|---|---|---|
| Advanced | NFS | 0.743 (0.718–0.767) | 0.750 (0.718–0.783) | 0.733 (0.695–0.770) | 0.505 |
| fibrosis | FIB4 | 0.779 (0.756–0.802) | 0.791 (0.761–0.821) | 0.763 (0.727–0.799) | 0.232 |
| F34 | FibroMeter[V3G] | 0.792 (0.769–0.815) | 0.789 (0.758–0.820) | 0.795 (0.761–0.830) | 0.797 |
| | CirrhoMeter[V3G] | 0.758 (0.732–0.783) | 0.760 (0.727–0.793) | 0.753 (0.715–0.792) | 0.790 |
| | VCTE | 0.820 (0.799–0.841) | 0.816 (0.787–0.845) | 0.829 (0.798–0.860) | 0.553 |
| | Agile3+ | 0.852 (0.832–0.871) | 0.846 (0.820–0.872) | 0.860 (0.831–0.888) | 0.464 |
| | Agile 4 | 0.838 (0.841–0.879) | 0.832 (0.804–0.860) | 0.845 (0.815–0.875) | 0.526 |
| Cirrhosis | NFS | 0.767 (0.734–0.801) | 0.766 (0.723–0.809) | 0.774 (0.723–0.826) | 0.814 |
| F4 | FIB4 | 0.816 (0.785–0.848) | 0.818 (0.781–0.856) | 0.814 (0.758–0.870) | 0.908 |
| | FibroMeter[V3G] | 0.820 (0.789–0.851) | 0.813 (0.774–0.853) | 0.832 (0.783–0.880) | 0.553 |
| | CirrhoMeter[V3G] | 0.812 (0.778–0.846) | 0.801 (0.756–0.846) | 0.830 (0.778–0.881) | 0.403 |
| | VCTE | 0.870 (0.845–0.895) | 0.859 (0.825–0.893) | 0.900 (0.870–0.931) | 0.079 |
| | Agile3+ | 0.893 (0.871–0.915) | 0.875 (0.844–0.905) | 0.925 (0.898–0.952) | 0.019 |
| | Agile 4 | 0.893 (0.869–0.917) | 0.875 (0.841–0.908) | 0.925 (0.897–0.953) | 0.023 |

NFS NAFLD fibrosis score, VCTE Vibration controlled transient elastography.
AUROC were compared using the two-sided Delong test.

3 + ≥ 0.679 with Agile4 ≤ 0.474; and (iv) Agile3 + ≥0.679 with Agile4 > 0.474. Similar results were obtained with FMV3G and CMV3G, with CMV3G also stratifying the risk of cirrhosis within the higher interval of FMV3G (Fig. 1c), leading to the four groups of the FM/CM classification (Fig. 1d): (i) FMV3G < 0.31; (ii) FMV3G between 0.31 and 0.76; (iii) FMV3G > 0.76 with CMV3G ≤ 0.40; and (iv) FMV3G > 0.76 with CMV3G > 0.40.

In the validation set, the prevalence of cirrhosis was less than 1% in the first group of Agile3 + /4 and FM/CM classifications (Fig. 2a, b). In their fourth group, the prevalence of cirrhosis was 59% for the Agile3 + /4 classification and 54% for the FM/CM classification.

**Study algorithm**
Based on the previous results, we selected the best candidate among simple blood tests (FIB4), specialized blood tests (FMV3G/CMV3G combination), and elastography-based tests (Agile3 + /4 combination). Multivariate analysis including these three candidates showed that Agile3 + /Agile4 and FMV3G/CMV3G combinations were both independent predictors of cirrhosis with no significant effect of FIB4. The latest guidelines from EASL propose the use of specialized blood tests to confirm the diagnosis of liver fibrosis made with VCTE[6]. We therefore evaluated whether the FM/CM classification (based on specialized blood tests) helps to refine the diagnosis of liver fibrosis made by the elastography-based Agile3 + /4 classification. Crossing of the two Agile3 + /4 and FM/CM classifications in the derivation set is depicted in the Supplementary Fig. s6. The prevalence of fibrosis stages in the 16 subgroups obtained led us to a final stratification of the patients in four diagnoses (Fig. 3a): F0-2; Liver biopsy required (grey zone); Advanced fibrosis F34; and Cirrhosis F4. Figure 3b (derivation set) and Fig. 3c (validation set) show fibrosis stages as a function of these four groups. In the validation set, 71% of the patients from the F4 group had confirmed cirrhosis on liver biopsy. Among cirrhotic patients, 47% were in the F4 group, 43% in the F34 group, 7% in the Biopsy group, and only 3% were misclassified in the F0-2 group. The study algorithm correctly diagnosed 86% of the patients and required liver biopsy in only 20% of the patients. Importantly, 88% of the misclassified patients in the validation set were by only one fibrosis stage (Supplementary Table s4). All these results were robust and did not differ between the derivation and the validation set.

Compared with the EASL pathway (Supplementary Fig. s7), the Agile3 + /4 classification and the study algorithm included more patients in the "F0-2" category while maintaining a high 85-90% diagnostic accuracy in this category (Supplementary Table s5). The study

algorithm outperformed the Agile3 + /4 classification in terms of patients correctly classified in the F34 and F4 categories (i.e., positive predictive value), as well as in the whole population. Finally, as compared to the currently recommended EASL pathway, the study algorithm maintained a high diagnostic accuracy in the validation set (respectively 86.1% vs 85.9%, p = 1.000), while providing a Cirrhosis-F4 category without any additional biomarker required.

We then focused on the accuracy of the study algorithm for the binary diagnosis of cirrhosis as compared to the Agile3 + /4 classification. For this analysis, we considered the categories F0-2 and Grey zone of these algorithms as the rule-out zone for cirrhosis, the F34 category as the undetermined zone for cirrhosis (no discrimination between F4 and F3 patients), and the F4 category as the rule-in zone. As compared to the Agile3 + /4 classification in the validation set, the study algorithm provided 20% increase in diagnostic accuracy in the rule-in zone (positive predictive value from 58.9% to 71.1%, Supplementary Table s6) and therefore less false positive results for cirrhosis.

Liver fibrosis measured by morphometry on biopsies from the Angers centre correlated well with fibrosis stages, especially the area of fibrosis (Rs = 0.548) and even more the area of portal fibrosis (Rs = 0.711, Supplementary Fig. s8). The area of fibrosis and the area of portal fibrosis progressively increased across the four groups of our study algorithm, the difference being significant between all pairs of adjacent groups (Supplementary Fig. s9a, b).

**Risk prediction charts**
Two risk prediction charts were developed in the derivation set, one for the diagnosis of cirrhosis and the other for advanced liver fibrosis. These charts were represented using contour plots showing the joint effect of Agile3 + /Agile4 and FMV3G/CMV3G on the risk of cirrhosis (Fig. 4a) and the risk of advanced liver fibrosis (Fig. 4b). As an example, for a patient with Agile3+ at 0.85, Agile4 at 0.68, FMV3G at 0.91 and CMV3G at 0.40, the predicted risk of advanced liver fibrosis is > 90% (Fig. 4b) and the risk of cirrhosis is 60–80% (Fig. 4a).

Figure 5a shows the calibration plot of the cirrhosis risk chart in the validation set. Calibration between the predicted risk by the chart and the observed prevalence of cirrhosis was excellent. The mean difference between the predicted risk by the cirrhosis risk chart (solid black line) and the perfect prediction (dotted blue line) was only −2.9% (extremes: −5% and −2%). We also evaluated the prevalence of cirrhosis in the 7 groups delineated by the cirrhosis risk chart. As expected, following the calibration plot analysis, the observed prevalence of cirrhosis matched very well with the predicted prevalence in each

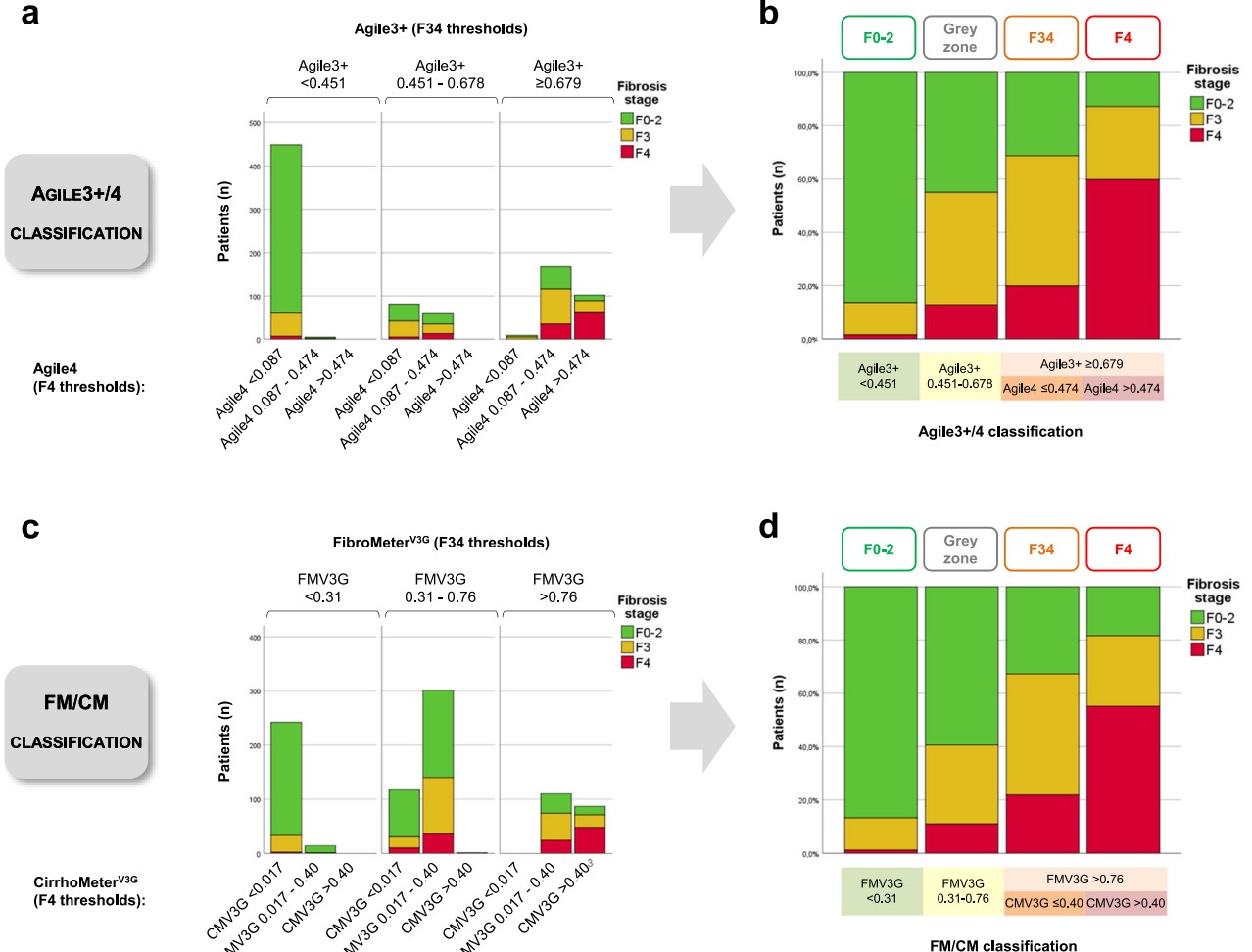

**Fig. 1 | Agile3 + /4 and FibroMeter^V3G/CirrhoMeter^V3G (FM/CM) classifications in the derivation set. a** Fibrosis stages as a function of subgroups defined by the F34 thresholds of Agile3+ and F4 thresholds of Agile4. **b** Fibrosis stage as a function of the four groups of the Agile3 + /4 classification. The four groups of the Agile3 + /4 classification result from the crossing of Agile3+ and Agile4 results as follows: (i) Agile3 + < 0.451; (ii) Agile3+ between 0.451-0.678; (iii) Agile 3 + ≥ 0.679 with Agile4 ≤ 0.474; and (iv) Agile3 + ≥0.679 with Agile4 > 0.474. **c** Fibrosis stages as a function of subgroups defined by the F34 thresholds of FibroMeter^V3G (FMV3G) and F4 thresholds of CirrhoMeter^V3G (CMV3G). **d** Fibrosis stage as a function of the four groups of the FM/CM classification. The four groups of the FM/CM classification result from the crossing of FMV3G and CMV3G results as follows: (i) FMV3G < 0.31; (ii) FMV3G between 0.31-0.76; (iii) FMV3G > 0.76 with CMV3G ≤ 0.40; and (iv) FMV3G > 0.76 with CMV3G > 0.40. Source data are provided as a Source Data file.

group (Fig. 5c), demonstrating the relevance of the cirrhosis risk chart for clinical practice. Our risk prediction chart improved the prediction of cirrhosis as compared to Agile4 or CMV3G alone, high results with these tests being associated with up to 25% overestimation of the probability of cirrhosis (Supplementary Fig. s10).

Same analysis was done for advanced fibrosis in the validation set. Calibration between the prediction of the advanced fibrosis risk chart and the observed prevalence of advanced fibrosis was excellent (Fig. 5b). The mean difference between the predicted risk and perfect prediction was only −2.7% (extremes: −5% and +2%). The observed prevalence of advanced fibrosis in the 10 groups delineated by the advanced fibrosis risk chart matched very well with the predicted prevalence (Fig. 5d).

Finally, morphometry analysis showed that area of fibrosis progressively increased with the risk of cirrhosis and the risk of advanced fibrosis as predicted by our risk charts (Supplementary Fig. s9c, d).

## Discussion

Previous studies on the non-invasive diagnosis of cirrhosis in NAFLD focused on the accuracy of single tests, with no study of tests in combination[13,14]. Moreover, a recent study has shown that non-invasive

tests are accurate in excluding cirrhosis, but they are much less accurate in confirming the diagnosis[15]. Innovative methods of biomarkers combination by machine learning have recently been tested but, compared to fibrosis tests available in clinical practice, they did not significantly increased AUROC for cirrhosis and they did not improve the ability to affirm cirrhosis with a poor positive predictive value between 40 and 60%[16]. We report here an approach that combines non-invasive fibrosis tests to accurately determine the presence or absence of cirrhosis in clinical practice as well as providing innovative risk prediction charts to allow for a personalized assessment of the individual probability of advanced liver fibrosis and cirrhosis.

Our study highlights the interest of combining the same sets of biomarkers in different equations designed for specific diagnostic targets. In this setting, the elastography-based tests Agile3+ and Agile4 both include VCTE results with simple blood markers and clinical parameters, the first test being dedicated to the diagnosis of advanced liver fibrosis and the second to cirrhosis. We found that cirrhosis is better excluded using the rule-out threshold of the test for advanced fibrosis (Agile3 + ) and better affirmed using the rule-in threshold of the test for cirrhosis (Agile4). Same results were obtained with the specialized blood tests FMV3G (dedicated to advanced fibrosis) and

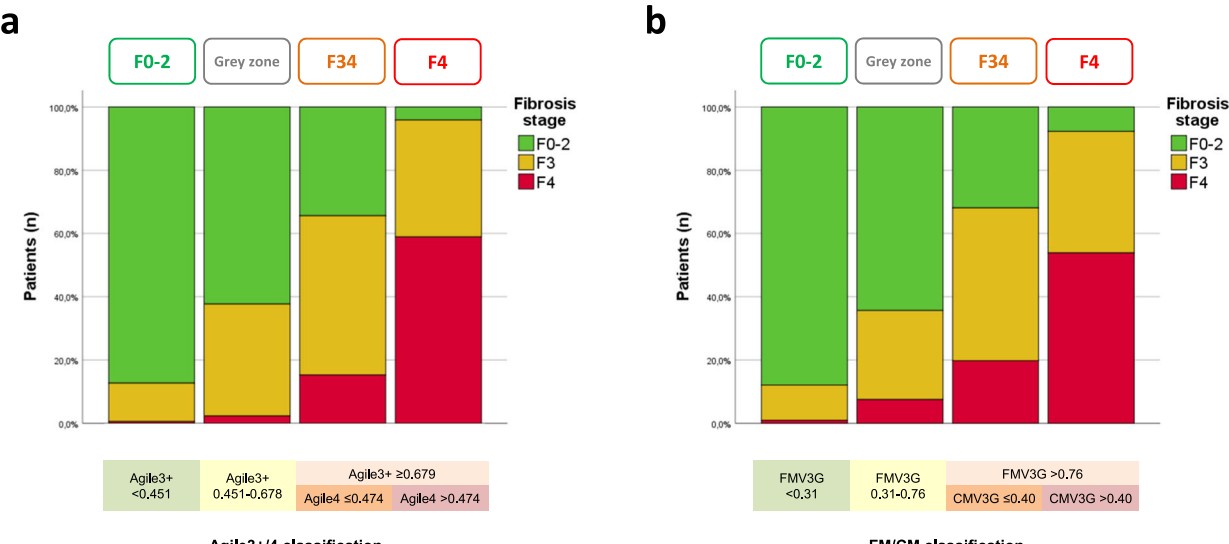

**Fig. 2 | Agile3+/4 and FibroMeterV3G/CirrhoMeterV3G (FM/CM) classifications in the validation set.** Fibrosis stages as a function of the four groups of the Agile3+/4 classification (**a**) and the four groups of the FM/CM classification (**b**) in the validation set. Source data are provided as a Source Data file.

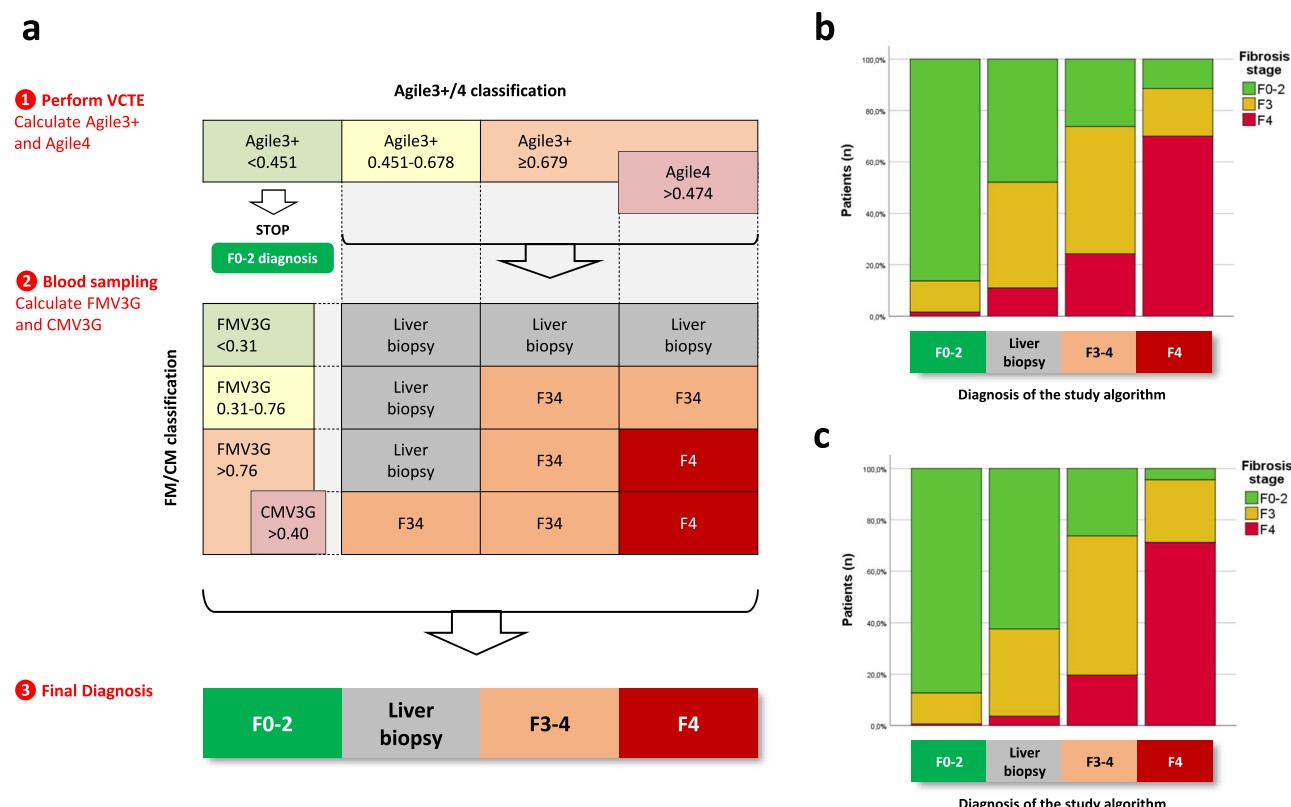

**Fig. 3 | Study algorithm. a** Algorithm description. The sequence of use of non-invasive tests is the same than recently recommended by the European Association for the Study of the Liver: first VCTE and second specialized blood test[6]. Specialized blood testing is performed after VCTE only if Agile3 + ≥0.451 to confirm the diagnosis in case of suspicion of advanced fibrosis/cirrhosis. The study algorithm stratifies patients in four diagnostic groups: F0-2 (no/mild fibrosis), Biopsy (i.e. undetermined diagnosis with liver biopsy required), F34 (advanced fibrosis), and F4 (cirrhosis). **b** Fibrosis stages as a function of the four groups defined by the study algorithm in the derivation set. **c** Fibrosis stages as a function of the four groups defined by the study algorithm in the validation set. CMV3G: CirrhoMeterV3G; FMV3G: FibroMeterV3G; FM/CM classification: FibroMeterV3G/CirrhoMeterV3G classification; VCTE: vibration controlled transient elastography. Source data are provided as a Source Data file.

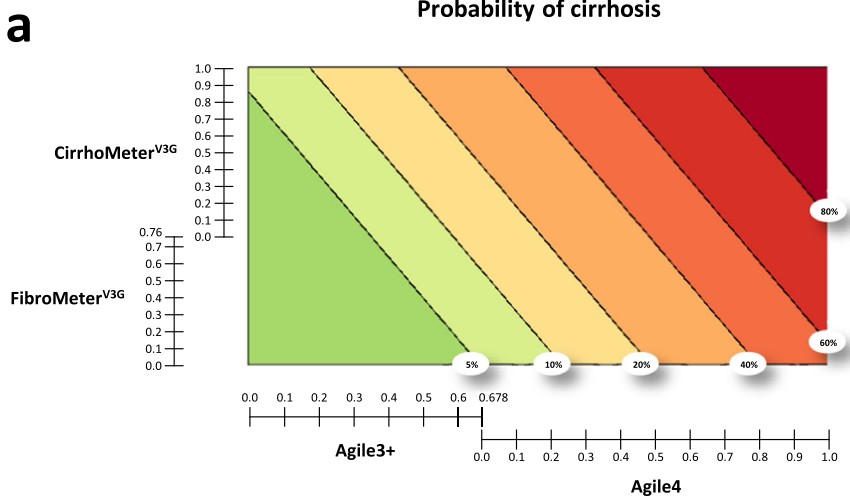

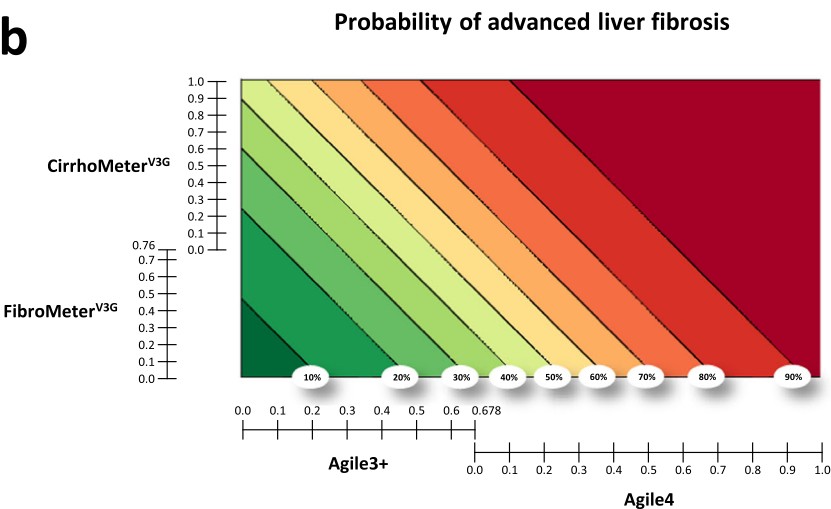

**Fig. 4 | Risk prediction charts.** Risk charts for the prediction of cirrhosis (**a**) and advanced fibrosis (**b**). When Agile3 + >0.678, move to the Agile4 scale; when FibroMeter^V3G > 0.76, move to the CirrhoMeter^V3G scale.

CMV3G (dedicated to cirrhosis). These findings led us to propose the Agile3 + /4 and FM/CM fibrosis classifications. Compared to single tests, these two classifications allowed the stratification of patients into four groups, with a low rate of liver biopsy required, whilst also allowing the synergistic advantage of composite tests with a very low rate of cirrhosis in the first exclusion group, and a last group enriched for cirrhosis.

Accumulated evidence in the field of liver fibrosis diagnosis shows that sequential combination of non-invasive tests, especially from different modalities (biology and elastography), improves the diagnostic accuracy. In line with these findings, we found that combining the elastography-based Agile3 + /4 and the blood-based FM/CM classifications improved the ability to affirm the diagnosis of cirrhosis with 71% prevalence in the last group (Fig. 3c). Our study algorithm (Fig. 3a) does not require additional measurements to those used in the latest EASL guidelines[6]. Indeed, in addition to VCTE, Agile3+ and Agile4 include only common blood markers (serum transaminases, platelets). Second, specialized blood testing is not required if the Agile3 + /4 classification diagnosis is F0-2. Third, specialized blood tests required after the Agile3 + /4 classification include the FibroMeter (recommended by the EASL following VCTE examination) and the CirrhoMeter (calculated from the same variables than the FibroMeter). Ultimately, our study algorithm can be considered as an improvement

and extension of the EASL algorithm, using the same parameters and sequence of fibrosis tests but providing a more precise and more accurate diagnosis of liver fibrosis.

Despite all these significant improvements, 43% of F4 patients from the validation set were included in the F34 group of our study algorithm, and therefore could not be differentiated from F3 patients. Such subgroup of difficult to diagnose cirrhosis was expected as no fibrosis tests could clearly delineate cirrhotic and non-cirrhotic patients. Because the probability of cirrhosis progressively increased with the result for each fibrosis tests, we moved from the classical approach consisting of a sequential use of fibrosis tests interpreted with diagnostic thresholds and providing a final semi-quantitative classification (Fig. 3a) to a more quantitative approach consisting of a personalized assessment of the individual probability of cirrhosis from the fibrosis tests results (Fig. 4a). The risk prediction charts we developed have the advantage of allowing the clinician to choose his level of requirement which may vary according to the clinical situation. As an example, a physician will require a very low risk to exclude cirrhosis in a patient. On the other hand, to include a patient in a clinical trial, a physician will require an acceptable rate of screen failure, as an example 30% (i.e. a 70% probability of cirrhosis). Such variability in choice is not possible in a sequential algorithm where test thresholds are fixed, which underlines the high interest of our risk

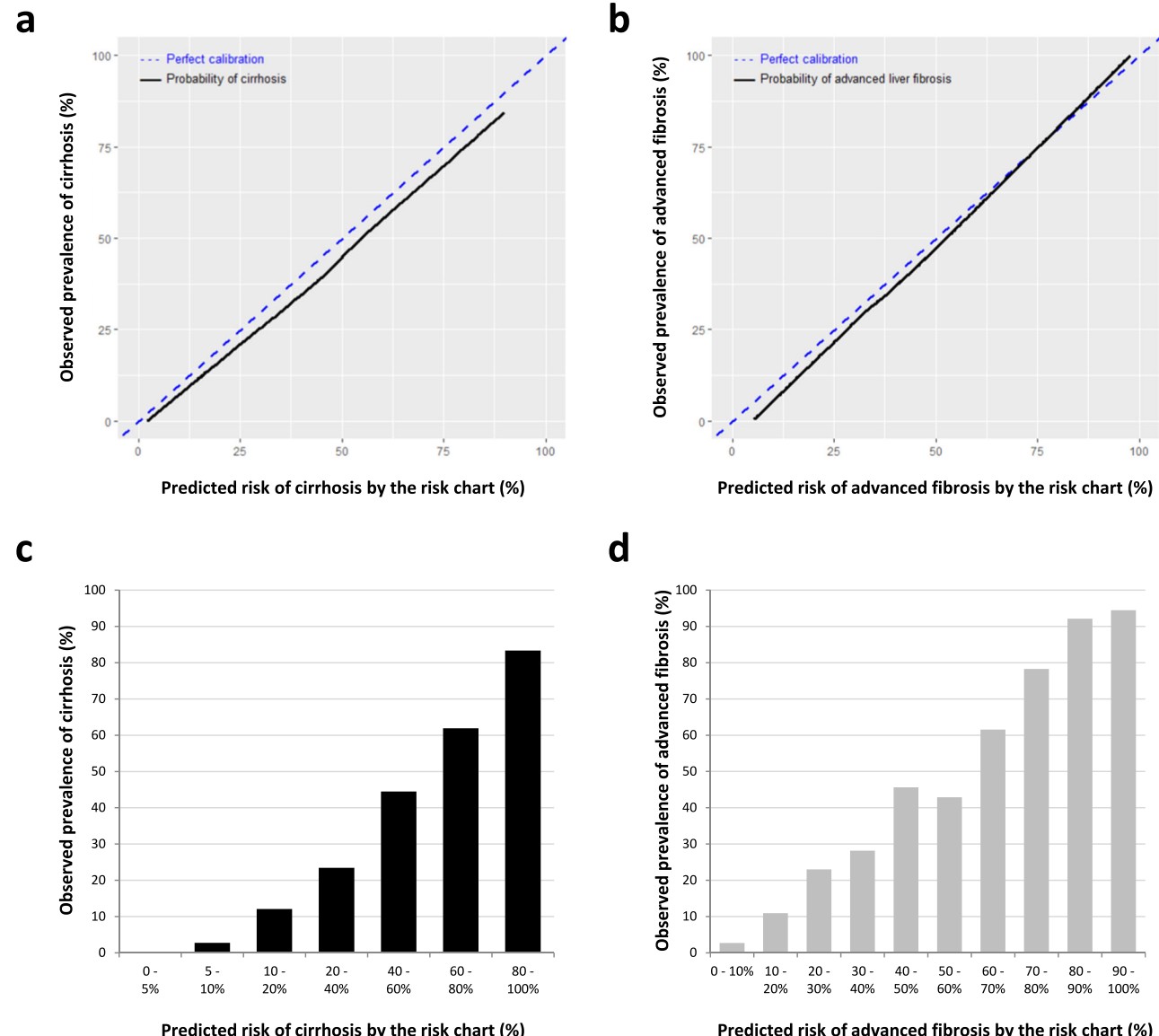

**Fig. 5 | Calibration of the predicted risk by the study risk charts in the validation set. a** Calibration plot of the predicted risk of cirrhosis by the risk chart presented in Fig. 4a. The blue dotted line represents perfect prediction (predicted risk of cirrhosis = observed prevalence of cirrhosis). The black solid line represents the observed prevalence of cirrhosis as a function of the predicted risk by the cirrhosis risk chart. **b** Calibration plot of the predicted risk of advanced fibrosis by the risk chart presented in Fig. 4b. **c** Patients were spitted into the 7 groups delineated by the cirrhosis risk chart, and the observed prevalence of cirrhosis is presented for each of these 7 groups. **d** Patients were spitted into the 10 groups delineated by the advanced fibrosis risk chart, and the observed prevalence of advanced fibrosis is presented for each of these 10 groups. Source data are provided as a Source Data file.

prediction charts for clinical practice. Finally, this approach based on risk assessment is in line with the personalized medicine developed in other medical specialties, such as the prediction of the cardiovascular risk[17].

Staging by a pathologist is a widely used method for the histological assessment of liver fibrosis, but morphometry provides a more quantitative, sensitive, and reproducible evaluation[18]. Morphometry has the advantage of highlighting significant effects unravelled by fibrosis staging in clinical trials[19]. Moreover, longitudinal studies have demonstrated that the area of fibrosis measured by morphometry is a better prognostic marker than fibrosis staging in chronic liver diseases[20,21]. Using this accurate method, we found that our study algorithm discriminates four groups with significant different amount of liver fibrosis. Moreover, the area of fibrosis progressively increased with the risk assessed by our prediction charts, therefore validating the relevance of such probabilistic approach for clinical practice. Two

interests of the quantitative scale provided by our risk prediction charts is that they will allow a precise delineation of the population at risk of liver-related complications as well as a fine evaluation of the cost-effectiveness threshold of specialized management. Further longitudinal works are now required to evaluate the prognostic significance of the disease severity stratification and the cost-effectiveness of the disease management using our new tools.

The strengths of our work are the very large number of patients included ($n = 1568$); the availability of several fibrosis tests (simple and specialized) utilizing different modalities (biology, elastography) and aimed at different diagnostic target (advanced liver fibrosis, cirrhosis); the multicentre phase 3 design with independent derivation and validation sets, which provides the highest evidence according to TRIPOD recommendations[22]; the robustness of the developed tools whose accuracy observed in the derivation set was confirmed in the validation set at each step of their development; and finally the opportunity to

validate our findings with liver morphometry, a quantitative method for liver fibrosis measurement which is more sensitive than the classical semi-quantitative staging on liver biopsy. Some patient characteristics were different between the derivation and the validation sets, reflecting the difference in practice across centres for patient recruitment. Nevertheless, the results we obtained did not significantly differ between the two sets, demonstrating the robustness of our study findings.

We acknowledge our study has some limitations. ELF is an accurate non-invasive test also proposed in the recommended algorithms for non-invasive diagnosis of liver fibrosis in NAFLD[6,23]. Unfortunately, ELF was not available and could not be tested in our study. Further work is needed to evaluate the interest of ELF in test combinations for the diagnosis of cirrhosis. Other devices besides blood tests and elastography are available in clinical practice to non-invasively assess cirrhosis such as Doppler ultrasonography, but this data was not available in our study. Whether Doppler ultrasonography helps to refine the diagnosis of cirrhosis for patients classified F34 by our study algorithm needs to be evaluated. Consensus reading of liver biopsies was performed in the UK cohort (PB, VP), but histological reading in the French centres was performed by the local pathologist (SM, BL, NS). Nevertheless, these pathologists were all experts specialized in hepatology with at least 20 years of experience. Previous studies have shown that inter-observer reproducibility for fibrosis staging is much better, even excellent, when performed by experts[24–26]. Our study algorithm and risk prediction charts have been developed in datasets coming from tertiary care centres and are therefore well suited for use in the context of liver clinics. With the growing awareness of the disease and further widespread use of non-invasive tests, one can expect our tools to disseminate to settings where the prevalence of cirrhosis is lower, such as diabetology clinics. Further work is needed to evaluate them in such context of use, and to determine whether they should be recalibrated for these settings.

In conclusion, the non-invasive diagnosis of cirrhosis remains an unmet need in NAFLD. The sequential algorithm and risk prediction charts we developed and validated represent a significant progress for the patient management in clinical practice. As they use the same fibrosis tests and sequence of tests use, these tools represent an extension and improvement of the current guidelines.

## Methods

### Patients

This study utilised four cohorts of adult patients with non-alcoholic fatty liver disease (NAFLD), liver biopsy, and vibration-controlled transient elastography (VCTE). Three of them were local and independent cohorts from three French University Hospitals (Angers, $n = 579$; Bordeaux, $n = 525$; and Grenoble, $n = 117$), as previously published in ref. 27 and updated for the present work. The fourth cohort ($n = 347$) came from a multicentre study performed in 7 liver centres across the United Kingdom[28]. All four cohorts obtained approval from Ethics Committees: CPP Ouest II Angers (CB2010-01) for Angers cohort; CPP Sud-Ouest et Outre Mer III for Bordeaux cohort; ARS Rhone Alpes (AC-2014-2094) for Grenoble cohort; and North Wales Research Ethics Committee (13/WA/0385) for UK cohort. IRB approval for the cohorts covered the work carried out here. All patients gave written informed consent before inclusion.

Patients included underwent liver biopsy as part of investigation of NAFLD after exclusion of concomitant steatosis-inducing drugs (such as corticosteroids, tamoxifen, amiodarone, or methotrexate), excessive alcohol consumption ( > 30 g/day in men or > 20 g/day in women), chronic hepatitis B or C infection, or evidence of other concomitant chronic liver disease. Patients were not included if they had presence or history of liver-related complication (ascites, variceal bleeding, jaundice, encephalopathy, hepatocellular carcinoma). Exclusion criteria for the present study were liver biopsy length

< 10 mm, VCTE failure, and missing blood markers for fibrosis tests calculation. All patients came from hepatology clinics and none of the biopsies were performed during bariatric surgery.

Data from the four cohorts were pooled in an excel file (Excel v2303, Microsoft, Redmond, WA, USA). The three French cohorts were not part of the development of the fibrosis tests evaluated in this study, whereas the UK cohort was part of the multicentre set in which Agile3+ and Agile4 fibrosis tests were developed. In addition, morphometry data (area of fibrosis) was only available for patients included from the Angers centre. We therefore decided to pool the UK and Bordeaux cohorts as a derivation set for our study, and the Angers and Grenoble cohorts as a validation set. This design allowed for: (i) a validation set that did not include any patient used for the derivation of the different fibrosis tests evaluated; (ii) a phase 3 design according to TRIPOD recommendations[22]; (iii) well-balanced derivation ($n = 872$) and validation ($n = 696$) sets; and (iv) the use of morphometry as an additional evaluation measure in the validation set. TRIPOD checklist is presented in Supplementary table s7.

### Liver histology

Pathological examinations were performed in each of the three French centres by a same senior expert specialized in hepatology (SM, BLB, NS) and blinded to patient data. We and others have shown excellent inter-observer reproducibility for liver fibrosis evaluation when performed by expert pathologists[24–26]. In the UK cohort, histological slides were analysed independently by two expert pathologists (PB, VP) who were blinded to each other's reading and to patient data. In case of disagreement, they reviewed the slides together to reach consensus. All experts involved in the study had at least 20 years of experience in liver histopathology. Liver fibrosis was evaluated according to the NASH CRN scoring system[24], i.e., F0: no fibrosis; F1: perisinusoidal or portal/periportal fibrosis, F2: perisinusoidal and portal/periportal fibrosis, F3: bridging fibrosis and F4: cirrhosis. Advanced liver fibrosis was defined as F3 + F4 fibrosis stages (F34).

For all patients included in Angers cohort, a picrosirius red-stained section of the whole liver biopsy was digitized in high-quality images (30,000 × 30,000 pixels, resolution of 0.5 μm/pixel). The area of whole fibrosis, the area of portal fibrosis and the area of perisinusoidal fibrosis were automatically measured on the digitized biopsy by morphometry software[29].

### Liver stiffness measurement

Liver stiffness measurement was performed using VCTE technology (FibroScan device; Echosens, Paris, France), by experienced operators and according to the manufacturer's recommendations. The measurements were performed in fasting conditions, within three months of the liver biopsy (within a week for 89% of the patients). The operators were blinded to histological and biological results. Agile3+ and Agile4 scores were calculated according to published formulas (Supplementary Table s1)[11].

### Blood fibrosis tests

Fasting blood samples were taken within a week of the liver biopsy. The following blood fibrosis tests were calculated according to published or patented formulas (Supplementary Table s1): FIB4, NAFLD fibrosis score, FibroMeter[V3G] (FMV3G), and CirrhoMeter[V3G] (CMV3G)[12,30,31]. CMV3G is a blood test that combines the same variables as the FMV3G, but which are combined in a formula with coefficients calculated specifically for the diagnosis of cirrhosis. All blood assays were performed in the laboratories of the investigating centres. We have previously demonstrated the excellent inter-laboratory reproducibility of blood fibrosis tests[32]. The diagnostic thresholds used for blood tests are detailed in Supplementary Table s1. When diagnostic thresholds were not available in the literature, we calculated them in the derivation set of the study, with further validation in the validation set.

## Statistics

Continuous variables were expressed as medians, with first and third quartiles, and compared using the Mann-Whitney or the Kruskal-Wallis tests. Categorical variables were expressed as percentages and compared using the Chi-squared or Fisher tests. Correlations between quantitative variables were determined using the Spearman correlation coefficient. The following analyses were performed in the derivation set and further validated in the validation set.

**Accuracy of non-invasive tests for the diagnosis of cirrhosis.** Diagnostic accuracy of the 7 non-invasive tests available (NAFLD fibrosis score, FIB4, FMV3G, CMV3G, VCTE, Agile3+ and Agile4) was evaluated using the area under the receiver operating characteristic (AUROC) and compared with the Delong test. Accuracy of the rule-out (sensitivity, negative predictive value) and the rule-in (specificity, positive predictive value) thresholds was also assessed.

**Complementary between non-invasive tests.** We then evaluate the complementarity between the tests developed for advanced fibrosis and those for cirrhosis. The aim was to evaluate how interact the tests including the same biomarkers but dedicated for different diagnostic targets (in our study: Agile3+ and Agile4, FibroMeter and CirrhoMeter), and whether their combination allows for a better patient stratification. If confirmed, such approach will help to improve the non-invasive diagnosis at the different steps of the recommended algorithms (elastography, specialized blood test).

**Agreement between non-invasive tests.** The best combination of tests for the diagnosis of cirrhosis was determined through multivariate binary logistic regression that included the best candidates identified from the previous results. The agreement between the selected tests was then assessed to produce the final diagnosis, resulting in a diagnostic algorithm including a cirrhosis category. Finally, our study algorithm was compared to the pathway proposed by the European Association for the Study of the Liver (Supplementary Fig. s7) to demonstrate the improvements made by our method.

**Risk prediction charts.** We finally developed a risk prediction chart that indicates the probability of cirrhosis (expressed as percentage) from the fibrosis tests results. This chart was represented using contour plots showing the joint effect of the fibrosis tests previously selected by the multivariate analysis. A risk prediction chart indicating the probability of advanced fibrosis was also developed.

Statistical analyses were performed using SPSS version 25.0 software (IBM, Armonk, NY, USA) and R version 3.6.2.

### Reporting summary

Further information on research design is available in the Nature Portfolio Reporting Summary linked to this article.

## Data availability

Data that support the findings of this study come from four independent cohorts in different countries (France, UK).

Angers cohort (France) – Individual deidentified participant data (including data dictionaries) that underlies the results reported in this article are available from the principal investigator of the cohort (Pr Jerome Boursier, JeBoursier@chu-angers.fr). No additional document will be shared. Data will be available immediately following publication, with no end date. Researchers who provide a methodologically sound proposal will be required to sign a data access agreement, and will only be allowed to carry out the objectives of the approved proposal.

Bordeaux cohort (France) – Individual deidentified participant data (including data dictionaries) that underlies the results reported in this article are available from the principal investigator of the cohort (Pr Victor de Ledinghen, victor.deledinghen@chu-bordeaux.fr). No additional document will be shared. Data will be available immediately following publication, with no end date. Researchers who provide a methodologically sound proposal will be required to sign a data access agreement, and will only be allowed to carry out the objectives of the approved proposal.

Grenoble cohort (France) – Individual deidentified participant data (including data dictionaries) that underlies the results reported in this article are available from the principal investigator of the cohort (Pr Charlotte Costentin, CCostentin@chu-grenoble.fr). No additional document will be shared. Data will be available immediately following publication, with no end date. Researchers who provide a methodologically sound proposal will be required to sign a data access agreement, and will only be allowed to carry out the objectives of the approved proposal.

UK cohort – Individual deidentified participant data (including data dictionaries) that underlies the results reported in this article are available from the principal investigator of the cohort (Pr Phil Newsome, p.n.newsome@bham.ac.uk). No additional document will be shared. Data will be available immediately following publication, with no end date. Researchers who provide a methodologically sound proposal will be required to sign a data access agreement, and will only be allowed to carry out the objectives of the approved proposal. Source data are provided with this paper.

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

## Acknowledgements

This work received grant support from Intercept, delivered to Angers University Hospital (JB). Philip N Newsome was supported by the Birmingham NIHR Biomedical Research Centre. This paper presents independent research supported in part by National institute for Health Research (NIHR) Birmingham Biomedical Research Centre at the University Hospitals Birmingham NHS foundation Trust and the University of Birmingham. (Grant reference number: BRC-1215-20009). The views expressed are those of author(s) and not necessarily those of NIHR or the department of Health and Social care. We thank Marc de Saint Loup, Sandra Girre, and Audrey Morisset for data collection and database construction.

## Author contributions

J.B. designed the study, supervised the statistical analysis, and drafted the manuscript. M.R. performed the statistical analysis. C.C., C.F.P., A.T., C.M.C., V.D.L., and P.N.N. critically reviewed the manuscript. J.C., S.M., B.L.B., V.P., P.B., and N.S. contributed to data acquisition in the study cohorts.

## Competing interests

Jerome Boursier: Board advisory and research grants to his institution from EchoSens. Other authors have nothing to declare.

## Additional information

## AFEF group for the study of liver fibrosis

**Justine Barthelon**[3,4], **Jerome Boursier**[1,2], **Paul Cales**[1,2], **Clémence Canivet**[1,2], **Julien Chaigneau**[2], **Charlotte Costentin**[3,4], **Thomas Decaens**[3,4], **Adèle Delamarre**[13], **Paul Hermabessiere**[13], **Marie Irles-Depé**[13], **Victor de Ledinghen**[9,13], **Marie-Noelle Hilleret**[3], **Isabelle Fouchard-Hubert**[1,2], **Adrien Lannes**[1,2], **Brigitte Le Bail**[8,9], **Sophie Michalak**[2,7], **Valérie Moal**[2,17], **Fréderic Oberti**[1,2], **Marine Roux**[2] **& Nathalie Sturm**[12]

[17]Biochemistry Department, Angers University Hospital, Angers, France.

## M118 study group

**Michael Allison**[18], **Quentin M. Anstee**[19], **Pierre Bedossa**[10,11], **Jeremy F. Cobbold**[20], **Jonathan J. Deeks**[21], **Peter J. Eddowes**[14,15,16,22], **Indra N. Guha**[22], **Philip N. Newsome** ⓘ [14,15,16], **Valérie Paradis**[10], **David Sheridan**[23] **& Emmanuel Tsochatzis**[24]

[18]Liver Unit, Addenbrooke's Hospital, Cambridge Biomedical Research Centre, Cambridge, UK. [19]Institute of Cellular Medicine, Faculty of Medical Sciences, Newcastle University, Newcastle upon Tyne, UK. [20]Department of Gastroenterology and Hepatology, Oxford University Hospitals NHS Foundation Trust, John Radcliffe Hospital, Oxford, UK. [21]National Institute for Health Research Biomedical Research Centre at University Hospitals Birmingham NHS Foundation Trust and the Institute of Applied Health Research, University of Birmingham, Birmingham, UK. [22]National Institute for Health Research Nottingham Biomedical Research Centre, Nottingham University Hospitals NHS Trust and University of Nottingham, Nottingham, UK. [23]Institute of Translational and Stratified Medicine, Faculty of Medicine and Dentistry, University of Plymouth, Plymouth, UK. [24]University College London Institute for Liver and Digestive Health, Royal Free Hospital, London, UK.

