## [Peer Review File · Nature Communications]

Practical diagnosis of cirrhosis in non-alcoholic fatty liver disease using currently available non-invasive fibrosis testsREVIEWER COMMENTS

Reviewer #1 (Remarks to the Author):

In this manuscript, Boursier et al utilize 4 separate cohorts and data from over 1500 patients with biopsy-proven NAFLD to develop practical rules on how to use NITs to diagnose NAFLD-induced cirrhosis. They developed an algorithm and a risk prediction chart to predict the presence of cirrhosis by liver specialists.

The topic is of great clinical significance and the manuscript highlights the low PPV of a single NIT to rule-in the presence of cirrhosis and the need to use combination NITs.

The scope of the manuscript might be a better fit for a subspecialty journal.

Several methodological issues limit my enthusiasm.

1. The variability in the pathologists' methods of reading the slides (1 pathologist at each center in the 3 French cohorts and consensus pathology reading in the English cohort). The authors should consider standardizing the way the slides are read by pathologists and using a consensus approach is highly encouraged.

2. Did the authors use any other clinical criteria to diagnose cirrhosis besides the pathologists readings? Any data on spleen size, collaterals, varices, or the presence of CSPH based on the Baveno criteria will be helpful. Unfortunately, some patients with what is labelled as F3 on liver biopsy will have cirrhosis based on a more comprehensive clinical assessment.

3. Given the additional cost associated with FibroMeter and CirrhoMeter, did the authors consider performing similar analysis using FIB4 (with cutoffs that correspond to F3-F4 and F4) as a second test after the AGILE3+/ AGILE4?

Reviewer #2 (Remarks to the Author):

In this manuscript, a large cohort of >1,500 NAFLD patients, from four different cohorts from France and England, with biopsy-based fibrosis staging combined with blood test and FibroScan analysis are analyzed for the prediction of advanced fibrosis and cirrhosis. After showing the good predictive capacity of both Fibroscan and blood test-based algorithms, they combined both methods to produce a new algorithm of stratification of the patients,

associated with different risks of advanced fibrosis and cirrhosis. The combination of both blood test algorithm and Elastometry based data are combined to produce a diagnosis of cirrhosis in large cohorts of patients with NAFLD, even if algorithm like Agile3+ and Agile4, used by the authors, already use blood test-based data to produce their results. Hence the study is clinically relevant and could be useful for guiding the clinicians in the future. Nevertheless, there are several limitations which should be addressed.

Major comments:

-The choice of the blood test algorithm should be better clarified and discussed. This algorithm was developed by the authors group and alternative blood test prediction method should be mentioned, like the ELF test. The choice should be justified based on comparative studies.

-The difference in age, cirrhosis rate, fibrosis stage and NAFLD activity score in between the training and the validation cohorts should be justified and discussed.

- The results should better mention the final sensitivity, specificity, positive and negative predictive values for the different predictions. In particular, for the prediction of cirrhosis. For now, authors show correlations between their scores and the level of fibrosis in patients, which is already well established. The purpose of this study is to combine different methods to produce a non-invasive diagnosis. The false discovery rate of this method should more clear and present in the abstract.

- As mentioned earlier, Agile3+ and Agile4 algorithm are already a combination of fibroscan and blood test results. In order to prove the benefit of their algorithm, authors should provide a comparison of their algorithm with Agile3+ and Agile4 alone or in combination, in terms of sensitivity, specificity, positive and negative predictive values.

- Even if the cohorts have been published elsewhere, it seems reasonable to provide a flow chart of patients' inclusion and exclusion criteria. Discuss possible sources of selection bias in the presented cohort.

- The paper lacks a statement on missing data and how they were handled
- More in general, the Authors declare compliance with TRIPOD recommendations. Please compile and provide the TRIPOD checklist as supplement
- Morphological ultrasound diagnosis of cirrhosis is as non-invasive and widely available as the proposed serum biomarkers. Given its high accuracy in the diagnosis of cirrhosis, its role should be discussed. If available, such data should be included in a comprehensive model.
- The total number of included patients is high, though cohort splitting is likely to have introduced inefficiencies below a critical threshold. Please discuss this limitation, or consider using bootstrapping or cross-validation.

Minor comments:

- The paper often refers to a group F34 (which I suppose to be F3+F4). Please define it clearly in the method section.
- Please report performance measures with 95% confidence intervals.

1. The variability in the pathologists' methods of reading the slides (1 pathologist at each center in the 3 French cohorts and consensus pathology reading in the English cohort). The authors should consider standardizing the way the slides are read by pathologists and using a consensus approach is highly encouraged.

Unfortunately, it was not possible to organize a central reading with consensus approach for the 1,568 liver biopsies of the study within the deadline of the reviewing process (the RV1 of the manuscript had to be produced within three months). We agree with the Reviewer that a consensus approach would be of particular interest for the diagnosis of NASH, as reproducibility between expert pathologists for this diagnosis is poor. However, several studies have shown that inter-observer reproducibility for fibrosis staging is much better, even excellent, when performed by experts (as already emphasized in the "Patients and Methods" section with references #26-28). In our study, all liver biopsies were read by expert pathologists who are all specialized in hepatology with at least 20 years of experience. Therefore, we respectfully believe that the histological staging of liver fibrosis in our work is accurate, although we have added a comment recognizing this point.

To better emphasize the experience of the pathologists involved in our study, we have added the following sentence in the paragraph "Liver histology" from the Methods section: *"All experts involved in the study had at least 20 years of experience in liver histopathology."*

Moreover, to acknowledge the point raised by the Reviewer, we have added the following sentences in the paragraph dedicated to the study limitations in the Discussion section: *"Consensus reading of liver biopsies was performed in the UK cohort (PB, VP), but histological reading in the French centers was performed by the local pathologist (SM, BLB, NS). Nevertheless, these pathologists were all experts specialized in hepatology with at least 20 years of experience. Previous studies have shown that inter-observer reproducibility for fibrosis staging is much better, even excellent, when performed by experts²⁴⁻²⁶."*

2. Did the authors use any other clinical criteria to diagnose cirrhosis besides the pathologists readings? Any data on spleen size, collaterals, varices, or the presence of CSPH based on the Baveno criteria will be helpful. Unfortunately, some patients with what is labelled as F3 on liver biopsy will have cirrhosis based on a more comprehensive clinical assessment.

This comment is in line with the comment #8 of the Reviewer #2. As they were available only for a minority of the patients included, we did not use the Doppler ultrasonography or the upper gastrointestinal endoscopy data to classify patients as cirrhotic or not.

Following the Reviewer comment, we have added the following sentences in the Discussion section of the manuscript: *"Other devices besides blood tests and elastography are available in clinical practice to non-invasively assess cirrhosis such as Doppler ultrasonography, but this data was not available in our study. Whether Doppler ultrasonography helps to refine the diagnosis of cirrhosis for patients classified "F34" by our study algorithm needs to be evaluated."*

We could not use the Baveno criteria to classify patients in our study because it would have introduced a bias. Indeed, VCTE and elastography-based tests cannot be evaluated against a reference that already uses the biomarker VCTE (such as the Baveno criteria).

3. Given the additional cost associated with FibroMeter and CirrhoMeter, did the authors consider performing similar analysis using FIB4 (with cutoffs that correspond to F3-F4 and F4) as a second test after the AGILE3+/ AGILE4?

Following the comment of the Reviewer, we have included additional results which illustrate our choice.

Based on the diagnostic accuracy results obtained for the 9 non-invasive tests available in our study (FIB4, NFS, FMV3G, CMV3G, VCTE, Agile3+, Agile4, FMV3G/CMV3G combination, and Agile3+/4 combination), we selected the best candidate among simple blood tests (FIB4), specialized blood tests (FMV3G/CMV3G combination), and elastography-based tests (Agile3+/4 combination). This selection was mandatory as it is not possible to introduce the same test several times in a multivariate analysis (such as Agile3+ and Agile4 with their combination Agile3+/4; or FMV3G and CMV3G with the FMV3G/CMV3G combination). The multivariate analysis including the three candidates showed that Agile3+/Agile4 and FMV3G/CMV3G combinations were both independent predictors of cirrhosis with no significant effect of FIB4, demonstrating the relevance of this combination.

To further address the Reviewer comment, we performed an additional multivariate analysis including only Agile3+/Agile4 combination and FIB4. Result showed that Agile3+/Agile4 combination was independently associated with cirrhosis but not FIB4, demonstrating the lack of added value of FIB4 when associated with Agile3+/Agile4 combination for the prediction of cirrhosis.

Finally, in a pragmatic way, the combination of Agile3+/Agile4 and FMV3G/CMV3G has the advantage of following and improving the EASL pathway based on the agreement between VCTE and a specialized blood test. Indeed, by using Agile 3+ and Agile4 instead of VCTE, and the CirrhoMeter in addition to the FibroMeter, we were able to include a new “F4” diagnostic category in the EASL algorithm without the need for any additional biomarker.

Following the Reviewer comment we added the following sentences:

- Methods section, paragraph “Statistics”: *“The best combination of tests for the diagnosis of cirrhosis was determined through multivariate binary logistic regression that included the best candidates identified from the previous results.”*
- Results section, paragraph “New study algorithm”: *“Based on the previous results, we selected the best candidate among simple blood tests (FIB4), specialized blood tests (FMV3G/CMV3G combination), and elastography-based tests (Agile3+/4 combination). Multivariate analysis including these three candidates showed that Agile3+/Agile4 and FMV3G/CMV3G combinations were both independent predictors of cirrhosis with no significant effect of FIB4.”*

As the multivariate analysis including only Agile3+/Agile4 combination and FIB4 do not change our study results and messages, we choose to not present this additional result in the revised version of the manuscript, but we may add it at the request of the Reviewer and/or the Editor.

MAJOR COMMENTS

1. The choice of the blood test algorithm should be better clarified and discussed. This algorithm was developed by the authors group and alternative blood test prediction method should be mentioned, like the ELF test. The choice should be justified based on comparative studies.

Following the comments of the Reviewer, we have now better presented the context and objective of our study; the study methodology; and the results illustrating how we chose our test combination. We have also discussed the ELF as alternative blood test.

The corresponding paragraphs have been added in the manuscript:

- i) Better presentation of the study context and our objective in the introduction of the manuscript:
New non-invasive tests specifically developed for the diagnosis of cirrhosis are now available. Agile4 combines VCTE results with routine parameters from liver evaluation (serum transaminases, platelets, with sex and diabetes) in a formula dedicated to cirrhosis ¹¹. CirrhoMeter has been developed to target cirrhosis by using the same biomarkers than those included in the specialized blood test FibroMeter ¹². These tests dedicated to the diagnosis of cirrhosis offer the opportunity to improve the currently recommended algorithms for the non-invasive diagnosis of advanced liver fibrosis. Indeed, calculated at the same time as the tests used in these algorithms without the need for additional biomarkers, they may contribute to discriminate a new group including cirrhotic patients.
In this work, we used the best performing non-invasive tests currently available to liver specialists to develop and validate an accurate diagnosis of cirrhosis in patients with NAFLD, which translated in a new "cirrhosis" category in the diagnostic algorithms currently recommended.
- ii) Detailed description of the study methodology in the Paragraph "Statistics" in the Methods section:

The following analyses were performed in the derivation set and further validated in the validation set.

Accuracy of non-invasive tests for the diagnosis of cirrhosis – *Diagnostic accuracy of the 7 non-invasive tests available (NFS, FIB4, FMV3G, CMV3G, VCTE, Agile3+ and Agile4) was evaluated using the area under the receiver operating characteristic (AUROC) and compared with the Delong test. Accuracy of the rule-out (sensitivity, negative predictive value) and the rule-in (specificity, positive predictive value) thresholds was also assessed.*

Complementary between non-invasive tests – *We then evaluate the complementarity between the tests developed for advanced fibrosis and those for cirrhosis. The aim was to evaluate how interact the tests including the same biomarkers but dedicated for different diagnostic targets (in our study: Agile3+ and Agile4, FibroMeter and CirrhoMeter), and whether their combination allows for a better patient stratification. If confirmed, such approach will help to improve the non-invasive diagnosis at the different steps of the recommended algorithms (elastography, specialized blood test).*

Agreement between non-invasive tests – *The best combination of tests for the diagnosis of cirrhosis was determined through multivariate binary logistic regression that included the best candidates identified from the previous results. The agreement between the selected tests was*

then assessed to produce the final diagnosis, resulting in a new diagnostic algorithm including a “cirrhosis” category. Finally, our study algorithm was compared to the EASL pathway (Supplementary Figure s7) to demonstrate the improvements made by our method.

Risk prediction charts – We finally developed a risk prediction chart that indicates the probability of cirrhosis (expressed as percentage) from the fibrosis tests results. This chart was represented using contour plots showing the joint effect of the fibrosis tests previously selected by the multivariate analysis. A risk prediction chart indicating the probability of advanced fibrosis was also developed.

iii) Results illustrating how we chose our test combination in the paragraph “New study algorithm” in the Results section:

Based on the previous results, we selected the best candidate among simple blood tests (FIB4), specialized blood tests (FMV3G/CMV3G combination), and elastography-based tests (Agile3+/4 combination). Multivariate analysis including these three candidates showed that Agile3+/Agile4 and FMV3G/CMV3G combinations were both independent predictors of cirrhosis with no significant effect of FIB4.

iv) Discussion about ELF in the paragraph dedicated to the study limitations in the Discussion section:

We acknowledge our study has some limitations. ELF is an accurate non-invasive test also proposed in the recommended algorithms for non-invasive diagnosis of liver fibrosis in NAFLD ^{6,22}. Unfortunately, ELF was not available and could not be tested in our study. Further work is needed to evaluate the interest of ELF in test combinations for the diagnosis of cirrhosis.

2. The difference in age, cirrhosis rate, fibrosis stage and NAFLD activity score in between the training and the validation cohorts should be justified and discussed.

The difference in some characteristics observed between the training and the validation cohorts reflect the difference in practice across centres for the patient recruitment. Interestingly, despite these differences, the study results were not significantly different between the derivation and the validation sets demonstrating the robustness of our study findings.

Following the Reviewer comment, this relevant point was added to the discussion: “*Some patient characteristics were different between the derivation and the validation sets, reflecting the difference in practice across centres for patient recruitment. Nevertheless, the results we obtained did not significantly differ between the two sets, demonstrating the robustness of our study findings.*”

3. The results should better mention the final sensitivity, specificity, positive and negative predictive values for the different predictions. In particular, for the prediction of cirrhosis. For now, authors show correlations between their scores and the level of fibrosis in patients, which is already well established. The purpose of this study is to combine different methods to produce a non-invasive diagnosis. The false discovery rate of this method should more clear and present in the abstract.

4. As mentioned earlier, Agile3+ and Agile4 algorithm are already a combination of fibroscan and blood test results. In order to prove the benefit of their algorithm, authors should provide a comparison of their algorithm with Agile3+ and Agile4 alone or in combination, in terms of sensitivity, specificity, positive and negative predictive values.

We provide here answer to both comments #3 and #4. To address these comments, we have added new results to the manuscript.

We have compared our new study algorithm (Figure 2) with the Agile3+/4 classification (Figure 1) to show the interest of using the agreement between FMV3G/CM3VG and Agile3+/4 combinations as compared to Agile3+/4 combination alone. In the validation set, these results show better accuracy of our study algorithm in the “F34” and the “F4” category (i.e., positive predictive value), as well as in the whole set. We also compared our study algorithm to the EASL pathway based on the agreement between VCTE and a specialized blood test to demonstrate the interest of using the Agile3+/4 combination instead of VCTE alone and the FMV3G/CM3VG combination instead of FMV3G alone. As compared to EASL pathway our study algorithm maintained the diagnostic accuracy while having the high interest in adding a new “cirrhosis” category in the algorithm without any additional biomarker required.

The following paragraph has been added to the Result section of the manuscript: *“Compared with the EASL pathway (Supplementary Figure s7), the Agile3+/4 classification and the study algorithm included more patients in the “F0-2” category while maintaining a high 85-90% diagnostic accuracy in this category (Supplementary Table s5). The study algorithm outperformed the Agile3+/4 classification in terms of patients correctly classified in the “F34” and “F4” categories (i.e., positive predictive value), as well as in the whole population. Finally, as compared to the currently recommended EASL pathway, the study algorithm maintained a high diagnostic accuracy in the validation set (respectively 86.1% vs 85.9%, $p=1.000$), while providing a new “cirrhosis-F4” category without any additional biomarker required.”*

We then evaluated our study algorithm versus the Agile3+/4 classification alone for the binary diagnosis of cirrhosis. Results shows that our study algorithm provided a 20% increase in diagnostic accuracy in the rule-in zone (positive predictive value from 58.9% to 71.1%), and therefore less false positive results for cirrhosis.

The following paragraph has been added to the Result section of the manuscript: *“We then focused on the accuracy of the study algorithm for the binary diagnosis of cirrhosis as compared to the Agile3+/4 classification. For this analysis, we considered the categories “F0-2” and “Grey zone” of these algorithms as the rule-out zone for cirrhosis, the “F34” category as the undetermined zone for cirrhosis (no discrimination between F4 and F3 patients), and the “F4” category as the rule-in zone. As compared to the Agile3+/4 classification in the validation set, the study algorithm provided 20% increase in diagnostic accuracy in the rule-in zone (positive predictive value from 58.9% to 71.1%, Supplementary Table s6) and therefore less false positive results for cirrhosis.”*

Unfortunately, it was not possible to update the abstract as requested by the reviewer due to the very small number of words allowed which did not allow us to include more details (no more than 150 words for the abstract).

5. Even if the cohorts have been published elsewhere, it seems reasonable to provide a flow chart of patients' inclusion and exclusion criteria. Discuss possible sources of selection bias in the presented cohort.

We initially presented the patients finally selected for the study. Following the Review comment, we now present the flow chart from the data received to the selected derivation and validation sets (see new Supplementary Figure s1). The paragraphs “Patients” in the Methods section and the Results section have been updated accordingly.

About the possible sources of selection bias, we acknowledge that our patients were from tertiary care centers and that our study algorithm and risk prediction charts now need to be further validated in patients from different settings. This point was already addressed in the discussion of the manuscript.

6. The paper lacks a statement on missing data and how they were handled

As stated in the study flow chart now presented in the RV1 (see new Supplementary Figure s1), patients with missing data for VCTE or blood fibrosis tests were excluded from the study.

7. More in general, the Authors declare compliance with TRIPOD recommendations. Please compile and provide the TRIPOD checklist as supplement

Reference to TRIPOD recommendations (Collins et al, BMJ 2015) has been added in the paragraph "Patients" from the Methods section (ref #25). We also added the TRIPOD checklist in Supplementary Table s7, as requested by the Reviewer.

To follow TRIPOD recommendation, the title of the manuscript has been updated: *"Development and validation of a practical diagnosis of cirrhosis (...)"*

8. Morphological ultrasound diagnosis of cirrhosis is as non-invasive and widely available as the proposed serum biomarkers. Given its high accuracy in the diagnosis of cirrhosis, its role should be discussed. If available, such data should be included in a comprehensive model.

This comment is in line with the Comment #3 of the Reviewer #1. Unfortunately, Doppler ultrasonography data were available only for a minority of the patients and were thus not used in our study.

Following the Reviewer comment, we have added the following sentences in the Discussion section of the manuscript:

"Other devices besides blood tests and elastography are available in clinical practice to non-invasively assess cirrhosis such as Doppler ultrasonography, but this data was not available in our study. Whether Doppler ultrasonography helps to refine the diagnosis of cirrhosis for patients classified "F34" by our study algorithm needs to be evaluated."

9. The total number of included patients is high, though cohort splitting is likely to have introduced inefficiencies below a critical threshold. Please discuss this limitation, or consider using bootstrapping or cross-validation.

Developing a model with internal validation (bootstrapping or cross-validation) corresponds to a phase 1b study according to TRIPOD recommendations.

We choose our study design to be consistent with validation requirements. Indeed, external validation of a prognostic model is needed before it can be used in daily practice to guide patient care (PMID 19477892). Accordingly, authors for the TRIPOD statements stated that "After developing a prediction model, it is strongly recommended to evaluate the performance of the model in other participant data than were used for the model development" (PMID 25569120, ref #25 in the manuscript).

As answered to the comment #2 of the Reviewer, our study results were not significantly different between the derivation and the validation sets (Table s5). This shows the reproducibility and transportability of our study algorithm and the lack of need for adjustment. In other words, because the results are similar between the derivation and validation sets, developing our algorithm in the whole population would have produced the same results.

For all these reasons, we respectfully estimate that our study design (phase 3 according to TRIPOD recommendations with development and validation of the model on separate data) provides the best evidence for our study results.

To address the Reviewer comment, we have updated the following sentence in the paragraph dedicated to the strengths of the study in the Discussion section: *“the multicentre phase 3 design with independent derivation and validation sets, which provides the highest evidence according to TRIPOD recommendations²²”*.

The sentences previously added to this paragraph following the comment #2 also address the comment #9: *“Some patient characteristics were different between the derivation and the validation sets, reflecting the difference in practice across centres for patient recruitment. Nevertheless, the results we obtained did not significantly differ between the two sets, demonstrating the robustness of our study findings”*.

MINOR COMMENTS

10. The paper often refers to a group F34 (which I suppose to be F3+F4). Please define it clearly in the method section.

To make clarification, we have added the following sentence in the paragraph “Liver Histology” from the Methods section: *“Advanced liver fibrosis was defined as F3+F4 fibrosis stages (F34)”*.

11. Please report performance measures with 95% confidence intervals.

As required by the Reviewer, 95%CI are now presented in the Table 2 and in the Table included in the Supplementary Figure s8.

REVIEWERS' COMMENTS

Reviewer #1 (Remarks to the Author):

Although the authors were not able to address comments 1 and 2, I understand their reasons and appreciate their detailed response to comment 3.

Reviewer #2 (Remarks to the Author):

In the revised version, key comments have been largely addressed.